# 3-Benzoylisoxazolines by 1,3-Dipolar Cycloaddition: Chloramine-T-Catalyzed Condensation of α-Nitroketones with Dipolarophiles

**DOI:** 10.3390/molecules26123491

**Published:** 2021-06-08

**Authors:** Xinhui Pan, Xiaobing Xin, Ying Mao, Xin Li, Yanan Zhao, Yidi Liu, Ke Zhang, Xiaoda Yang, Jinhui Wang

**Affiliations:** 1Stake Key Laboratory of Natural and Biomimetic Drugs, Department of Chemical Biology, School of Pharmaceutical Sciences, Peking University, Beijing 100191, China; xyang@bjmu.edu.cn; 2Key Laboratory of Xinjiang Phytomedicine Resource and Utilisation, Ministry of Education, School of Pharmaceutical Sciences, Shihezi University, Shihezi 832002, China; 18699300391@163.com (X.X.); lixin@stu.shzu.edu.cn (X.L.); zynhyl@stu.shzu.edu.cn (Y.Z.); lyd@stu.shzu.edu.cn (Y.L.); xjzk1984@163.com (K.Z.); 3Department of Pharmacology, Houbo College of Xinjiang Medical University, Karamay 834000, China; 4School of Chemistry and Chemical Engineering, Shandong University, Jinan 250100, China; maoying1984@163.com

**Keywords:** isoxazolines, α-nitroketones, alkenes, chloramine-T, 1,3-dipolar cycloaddition

## Abstract

In this study, 3-benzoylisoxazolines were synthesized by reacting alkenes with various α-nitroketones using chloramine-T as the base. The scope of α-nitroketones and alkenes is extensive, including different alkenes and alkynes to form various isoxazolines and isoxazoles. The use of chloramine-T, as the low-cost, easily handled, moderate base for 1,3-dipolar cycloaddition is attractive.

## 1. Introduction

Isoxazoline derivatives have been demonstrated to exhibit a variety of biological and pharmacological activities, such as antithrombotic effects, insect growth regulation, immunopotentiation and anticancer activities (Figure 1) [1,2,3,4,5]. In organic synthesis, isoxazolines are also useful intermediates. For instance, they can be converted into different critical synthetic units, such as β-hydroxy ketones [6], γ-amino alcohols [7], α,β-unsaturated ketones [8] and β-hydroxy nitriles [9]. Carreira et al. and Lee et al. [10,11,12] used isoxazolines in the total synthesis of natural products (Figure 2).

Previous studies described various methods for synthesizing isoxazolines, namely, the 1,3-dipolar cycloaddition of dipolarophiles [13] (alkynes, alkenes) with nitrile oxides from aldoximes [14] or α-nitroketones. In the case of α-nitroketones, nitrile oxides are prepared by dehydration with an acid (such as sulfuric acid [15], *p*-toluenesulfonic acid [16,17] or polyphosphoric acid-silica (PPA/SiO_2_) [18]) or a base (such as N-methylimidazole [19], 1,4-diazabicyclo [2.2.2] octane [20] and copper (II) acetate/N-methylpiperidine [21]) as the catalytic systems. 

Moderate to good reaction efficiency can be obtained by utilizing these acidic or basic catalytic systems. The majority of the reactions undergo 1,3-dipolar cycloadditions with different α-nitroketones to obtain various isoxazoles. However, it is reported that 1,3-dipolar cycloadditions that start from nitroalkanes bearing a carbonyl group do not proceed smoothly and thus, generate the desired products in low yields. Tsubaki et al. [22] recently developed nitrile oxide cycloaddition reactions between nitrile oxides derived from O-alkyloxime-substituted nitroalkanes and various alkenes to construct 2-isoxazolines with electron-withdrawing groups. Although this method proceeds smoothly with nitroalkanes and various alkenes, it requires the preparation of O-alkyloxime-substituted nitroalkanes as precursors. To simplify this reaction, we tested several other bases and identified chloramine-T a commercially available agent, to give superior results [23]. We found that chloramine salts showed highly attractive practical characteristics: easy and amenable preparation at large scales, nontoxic by-products, excellent reactivity and high stability to air and heat. In this study, we successfully developed a chloramine-T catalytic system for 1,3-dipolar cycloaddition of dipolarophiles with α-nitroketones. To the best of our knowledge, it is the first time that chloramine-T has been used in 1,3-dipolar cycloaddition reactions to obtain isoxazolines from α-nitroketones. 

## 2. Results and Discussion

Benzoylnitromethane **1a** (1 equiv) was reacted with allylbenzene **2a** (5 equiv) in the presence of various bases in acetonitrile at 80 °C. The results are summarized in Table 1 (entries 1–6). No reaction occurred in the base-free system. Chloramine-T was the best among the bases tested herein. It is conceivable that 0.5 equivalents of chloramine-T can provide a better yield (Table 1, entries 6–9). According to the results, a few solvents were screened for optimization. Cycloaddition was found to be reliant on the solvent. Although H_2_O inhibited the cycloaddition, the reaction worked in DMSO and DMF, while CH_3_CN as the solvent provided the best results (Table 1, entries 10–13), producing **3a** with a yield of 77%. Increasing the temperature to 90 °C or lowering it to 60 °C resulted in a lesser yield (Table 1, entries 14 and 15). Thus, the reaction in optimal conditions was conducted at 80 °C for 18 h with 0.5 equiv of chloramine-T in the presence of acetonitrile as the solvent; and **3a** was obtained with a yield of 77%.

Under the optimized reaction conditions, various substrates were subjected to 1,3-dipolar cycloaddition (Scheme 1). Several electronically varied α-nitroketones were subjected to cycloaddition. Efficiency was considered as the sensitivity of cycloaddition to electronic substituents. Electron-deficient α-nitroketones (**3b**–**3d**, Scheme 1) provided products in slightly better yields related to electron-rich α-nitroketones (**3e** and **3f**, Scheme 1). Phenacyl nitro derivatives incorporating tert-butyl and phenyl at the para-position were also successful in the cycloaddition (**3g** and **3h**, Scheme 1). Simple nitroketones were efficiently coupled with allylbenzene (**3i**, Scheme 1).

Next, we investigated the scope of the alkenes in Scheme 2. Alkene alternative with contrasting and electronically varied substituents reacted with benzoylnitromethane **1a** smoothly under the standard conditions to obtain the desired products in excellent yields. Moreover, the electron-rich allylbenzenes and allylalkanes produced the product in good yields (**5a**–**5e**, Scheme 2). Similarly, the electron-deficient ones, such as allyl chloride, produced the product in excellent yields (**5f**, Scheme 2). Good yields were also obtained in the cycloaddition of cyclohexene (**5g**, Scheme 2).

Finally, **1a** was reacted with 1-hexyne in the presence of chloramine-T in acetonitrile at 70 °C for 18 h. Isoxazoles **7a** and **7b** were obtained with a yield of 68% and 64% (Scheme 3). This result demonstrated that this reaction is also suitable for cycloadditions to form isoxazoles.

A possible mechanism for the reaction is shown in Scheme 4. In acetonitrile, the ion pair **9**, which is formed between nitronate **1** and the protonated base **8**, undergoes cycloaddition with dipolarophile **2** to obtain intermediate **10**. Subsequently, the ion pair intermediate adduct **10** releases chloramine-T **8** to produce the 2-hydroxyoxazolidine **11**, which is then dehydrated to give the product **3**. Finally, another nitronate **1** reacts with chloramine-T **8** to obtain the new ion pair intermediate **9**.

## 3. Experimental Section

### 3.1. General Experimental Methods

The structures of produced compounds were firmly confirmed by ^13^C NMR and ^1^H NMR spectra and supported by HRMS, IR data (see the Appendix A). 

^1^H NMR (400 MHz) and ^13^C NMR (101 MHz) were recorded at room temperature by using a DRX-400 spectrometer (Bruker, Germany) in CDCl_3_. Chemical shifts were given in parts per million (ppm) on the delta (δ) scale. The solvent peak was used as a reference value, for ^1^H NMR: CDCl_3_ δ 7.26; for ^13^C NMR: CDCl_3_ at 77.16 ppm. IR spectra were recorded using an Avatar 360 FT-IR ESP spectrometer Nicolet (Waltham, MA, USA) at room temperature. HR-ESI-MS spectra were acquired using an Agilent 6210 ESI/TOF mass spectrometer (Agilent Technologies, Santa Clara, CA, USA). Analytical TLC was run on silica gel plates (GF254, Yantai Institute of Chemical Technology, Yantai, China). Spots on the plates were observed under UV light. Column chromatography was performed on silica gels (200~300 mesh and 300–400 mesh; Qingdao Marine Chemical Factory, Qingdao, China). Super-dry solvent CH_3_CN, DMSO and DMF were purchased from Aldrich and used as supplied. The α-nitroketones were synthesized using the same method as reported in the literature [16].

### 3.2. General Procedure for the Cycloaddition of Alkenes and α-Nitroketones

Chloramine-T (0.0625 mmol, 0.5 equiv) was added to a solution of **1** (0.125 mmol, 1 equiv) and **2** (0.625 mmol, 5 equiv) (or **4** (0.625 mmol, 5 equiv) or **6** (0.625 mmol, 5 equiv)) in CH_3_CN (0.2 mL). The mixture was then stirred at 80 °C until the starting material disappeared, as monitored by TLC. Subsequently, the mixture was directly purified by flash chromatography (with ethyl acetate/petroleum ether as the eluent) to obtain the desired product (**3**, **5** or **7**).

#### 3.2.1. (5-Benzyl-4,5-dihydroisoxazol-3-yl)(phenyl)methanone (**3a**)

The compound (with a yield of 77%) was prepared following the general procedure described in Section 3.2. ^1^H NMR (400 MHz, CDCl_3_) δ 8.22–8.12 (m, 2H), 7.63–7.57 (m, 1H), 7.51–7.44 (m, 2H), 7.38–7.33 (m, 2H), 7.28 (m, 3H), 5.08 (ddt, *J* = 10.8, 7.9, 6.3 Hz, 1H), 3.37 (dd, *J* = 17.6, 10.8 Hz, 1H), 3.19–3.08 (m, 2H), 2.98 (dd, *J* = 14.0, 6.5 Hz, 1H); ^13^C NMR (101 MHz, CDCl_3_) δ 186.2, 157.5, 135.8, 135.6, 133.3, 130.1(2C), 129.3(2C), 128.5(2C), 128.1(2C), 126.8, 83.3, 40.7, 38.1; IR νmax 3033, 1654, 1581, 710, 672 cm^−1^; HRMS (EI) *m*/*z* calcd for C_17_H_16_NO_2_ [M + H]^+^ 266.1176, found 266.1178. These data are consistent with the data reported in the literature [11].

#### 3.2.2. (5-Benzyl-4,5-dihydroisoxazol-3-yl)(2-bromophenyl)methanone (**3b**)

The compound was prepared following the general procedure. Yield 73%. ^1^H NMR (400 MHz, CDCl_3_) δ 7.51 (d, *J* = 7.6 Hz, 1H), 7.29–7.25 (m, 2H), 7.24 (m, 2H), 7.21 (m, 1H), 7.18 (m, 2H), 7.15 (m, 1H), 5.05 (ddt, *J* = 11.0, 7.6, 6.3 Hz, 1H), 3.22 (dd, *J* = 17.5, 10.9 Hz, 1H), 3.04–2.94 (m, 2H), 2.88 (dd, *J* = 14.0, 6.5 Hz, 1H); ^13^C NMR (101 MHz, CDCl3) δ 189.2, 158.1, 139.2, 135.8, 133.4, 132.1, 129.8, 129.7(2C), 128.8(2C), 127.2, 127.1, 120.0, 85.2, 41.0, 36.7; IR νmax 3034, 1680, 1585, 755, 694 cm^−1^; HRMS (EI) *m*/*z* calcd for C_17_H_15_NO_2_Br [M + H]^+^ 344.0281, found 344.0279.

#### 3.2.3. (5-Benzyl-4,5-dihydroisoxazol-3-yl)(3-bromophenyl)methanone (**3c**)

The compound was prepared following the general procedure. Yield 67%. ^1^H NMR (400 MHz, CDCl_3_) δ 8.24 (m, 1H), 8.07 (m, 1H), 7.70 (m, 1H), 7.37–7.33 (m, 2H), 7.32 (m, 1H), 7.27 (m, 3H), 5.16–5.03 (m, 1H), 3.34 (dd, *J* = 17.6, 10.9 Hz, 1H), 3.16–3.05 (m, 2H), 2.98 (dd, *J* = 14.0, 6.4 Hz, 1H); ^13^C NMR (101 MHz, CDCl3) δ 185.0, 157.6, 137.5, 136.4, 135.9, 133.2, 129.9, 129.6(2C), 128.9, 128.8(2C), 127.2, 122.6, 83.9, 40.9, 38.1; IR νmax 3038, 1691, 1616, 910, 811, 742 cm^−1^; HRMS (EI) *m*/*z* calcd for C_17_H_15_NO_2_Br [M + H]^+^ 344.0281, found 344.0293.

#### 3.2.4. (5-Benzyl-4,5-dihydroisoxazol-3-yl)(4-bromophenyl)methanone (**3d**)

The compound was prepared following the general procedure. Yield 72%. ^1^H NMR (400 MHz, CDCl_3_) ^1^H NMR (400 MHz, CDCl_3_) δ 8.04–7.97 (m, 2H), 7.62–7.56 (m, 2H), 7.35–7.30 (m, 2H), 7.25 (m, 3H), 5.07 (ddt, *J* = 10.9, 7.9, 6.3 Hz, 1H), 3.34 (dd, *J* = 17.6, 10.9 Hz, 1H), 3.13–3.05 (m, 2H), 2.96 (dd, *J* = 14.0, 6.5 Hz, 1H).; ^13^C NMR (101 MHz, CDCl_3_) δ 185.4, 157.9, 136.1, 134.6, 132.0(2C), 131.9(2C), 129.7(2C), 129.2, 128.9(2C), 127.2, 83.9, 41.1, 38.3; IR νmax 3046, 1696, 1615, 801, 750, 686 cm^−1^; HRMS (EI) *m*/*z* calcd for C_17_H_15_NO_2_Br [M + H]^+^ 344.0281, found 344.0276.

#### 3.2.5. (5-Benzyl-4,5-dihydroisoxazol-3-yl)(p-tolyl)methanone (**3e**)

The compound was prepared following the general procedure. Yield 68%. ^1^H NMR (400 MHz, CDCl_3_) δ 8.08–8.03 (m, 2H), 7.33 (m, 2H), 7.28 (m, 2H), 7.25 (m, 3H), 5.05 (ddt, *J* = 10.8, 7.9, 6.3 Hz, 1H), 3.35 (dd, *J* = 17.6, 10.8 Hz, 1H), 3.15–3.07 (m, 2H), 2.96 (dd, *J* = 14.0, 6.6 Hz, 1H), 2.42 (s, 3H); ^13^C NMR (101 MHz, CDCl3) δ 186.1, 157.9, 144.7, 136.2, 133.4, 130.6(2C), 129.6(2C), 129.2(2C), 128.8(2C), 127.1, 83.5, 41.1, 38.6, 21.9; IR νmax 3023, 2923, 1650, 1600, 831, 750, 693 cm^−1^; HRMS (EI) *m*/*z* calcd for C_18_H_18_NO_2_ [M + H]^+^ 280.1332, found 280.1336.

#### 3.2.6. (5-Benzyl-4,5-dihydroisoxazol-3-yl)(4-methoxyphenyl)methanone (**3f**) 

The compound was prepared following the general procedure. Yield 54%. ^1^H NMR (400 MHz, CDCl_3_) δ 8.23–8.15 (m, 2H), 7.36–7.30 (m, 2H), 7.29–7.25 (m, 3H), 6.97–6.91 (m, 2H), 5.04 (ddt, *J* = 10.8, 7.9, 6.4 Hz, 1H), 3.88 (s, 3H), 3.36 (dd, *J* = 17.6, 10.8 Hz, 1H), 3.16–3.07 (m, 2H), 2.96 (dd, *J* = 14.0, 6.6 Hz, 1H); ^13^C NMR (101 MHz, CDCl3) δ 184.7, 164.2, 157.9, 136.3, 132.9(2C), 129.6(2C), 128.8(2C), 127.1, 113.8(2C), 83.3, 55.7, 41.1, 38.8; IR νmax 3022, 2801, 1618, 1531, 802, 719, 676 cm^−1^; HRMS (EI) *m*/*z* calcd for C_18_H_18_NO_3_ [M + H]^+^ 296.1281, found 296.1285.

#### 3.2.7. (5-Benzyl-4,5-dihydroisoxazol-3-yl)(4-tert-butylphenyl)methanone (**3g**) 

The compound was prepared following the general procedure. Yield 63%. ^1^H NMR (400 MHz, CDCl_3_) δ 8.10–8.05 (m, 2H), 7.49–7.43 (m, 2H), 7.34–7.23 (m, 5H), 5.02 (ddt, *J* = 10.8, 7.9, 6.3 Hz, 1H), 3.32 (dd, *J* = 17.6, 10.8 Hz, 1H), 3.14–3.04 (m, 2H), 2.93 (dd, *J* = 14.0, 6.5 Hz, 1H), 1.34 (d, *J* = 4.4 Hz, 9H); ^13^C NMR (101 MHz, CDCl3) δ 186.0, 157.8, 157.4, 136.2, 133.3, 130.3(2C), 129.5(2C), 128.7(2C), 126.9, 125.4(2C), 83.4, 40.9, 38.4, 35.2, 31.1(3C); IR νmax 3030, 1660, 1601, 1403, 1375, 860, 750, 700 cm^−1^; HRMS (EI) *m*/*z* calcd for C_21_H_24_NO_2_ [M + H]^+^ 322.1802, found 322.1808.

#### 3.2.8. [1,1′-Biphenyl]-4-yl(5-benzyl-4,5-dihydroisoxazol-3-yl)methanone (**3h**)

The compound was prepared following the general procedure. Yield 57%. ^1^H NMR (400 MHz, CDCl_3_) δ 8.25–8.19 (m, 2H), 7.69–7.68 (m, 1H), 7.67 (m, 1H), 7.65 (m, 1H), 7.63 (m, 1H), 7.49 (m, 1H), 7.47 (m, 1H), 7.46 (m, 1H), 7.43 (m, 1H), 7.35–7.33 (m, 1H), 7.32 (m, 1H), 7.30–7.28 (m, 2H), 5.08 (ddt, *J* = 10.8, 7.9, 6.3 Hz, 1H), 3.38 (dd, *J* = 17.6, 10.8 Hz, 1H), 3.17–3.09 (m, 2H), 2.98 (dd, *J* = 14.0, 6.5 Hz, 1H); ^13^C NMR (101 MHz, CDCl3) δ 186.0, 157.9, 146.4, 139.9, 136.2, 134.6, 131.0(2C), 129.6(2C), 129.1(2C), 128.8(2C), 128.5(2C), 127.5(2C), 127.1(2C), 83.7, 41.1, 38.5; IR νmax 3029, 1655, 1648, 1401, 1362, 842, 746, 693 cm^−1^; HRMS (EI) *m*/*z* calcd for C_23_H_20_NO_2_ [M + H]^+^ 342.1489, found 342.1483. These data are consistent with the data reported in the literature [11].

#### 3.2.9. Methyl 5-benzyl-4,5-dihydroisoxazole-3-carboxylate (**3i**)

The compound was prepared following the general procedure. Yield 30%.^1^H NMR (400 MHz, CDCl_3_) δ 7.32 (m, 2H), 7.26 (m, 1H), 7.22 (m, 2H), 5.06 (ddd, *J* = 14.7, 10.9, 6.7 Hz, 1H), 3.86 (s, 3H), 3.18 (dd, *J* = 17.7, 10.9 Hz, 1H), 3.11 (dd, *J* = 14.0, 6.1 Hz, 1H), 2.94 (dd, *J* = 14.8, 5.3 Hz, 1H), 2.92–2.85 (m, 1H); ^13^C NMR (101 MHz, CDCl3) δ 161.2, 151.3, 136.0, 129.5(2C), 128.8(2C), 127.1, 84.5, 52.8, 40.8, 37.9; IR νmax 3418, 3032, 1717, 1584, 1449, 1366, 1265, 1126, 949, 746, 702, 582 cm^−1^; HRMS (EI) *m*/*z* calcd for C_12_H_14_NO_3_ [M + H]^+^ 220.0974, found 220.0980. 

#### 3.2.10. (5-(2-Methylbenzyl)-4,5-dihydroisoxazol-3-yl)(phenyl)methanone (**5a**) 

The compounds was prepared following the general procedure as isomers. Yield 71%. ^1^H NMR (400 MHz, CDCl_3_) δ 8.30–8.23 (m, 2H), 7.66 (m, 1H), 7.57–7.49 (m, 2H), 7.30–7.23 (m, 4H), 5.19–5.08 (m, 1H), 3.42 (ddd, *J* = 17.5, 10.7, 0.9 Hz, 1H), 3.22 (dt, *J* = 12.6, 7.5 Hz, 2H), 2.99 (dd, *J* = 14.3, 6.7 Hz, 1H), 2.44 (s, 3H); ^13^C NMR (101 MHz, CDCl3) δ 186.5, 157.9, 136.6, 135.9, 134.6, 133.6, 130.6, 130.4(2C), 130.0, 128.4(2C), 127.1, 126.3, 82.9, 38.6, 38.1, 19.8; IR νmax 3028, 2940, 1660, 1570, 750, 690 cm^−1^; HRMS (EI) *m*/*z* calcd for C_18_H_18_NO_2_ [M + H]^+^ 280.1332, found 280.1335.

#### 3.2.11. (5-(3-Methylbenzyl)-4,5-dihydroisoxazol-3-yl)(phenyl)methanone (**5b**) 

The compound was prepared following the general procedure. Yield 66%. ^1^H NMR (400 MHz, CDCl_3_) δ 8.15 (d, *J* = 7.2 Hz, 2H), 7.59 (m, 1H), 7.47 (m, 2H), 7.22 (m, 1H), 7.09 (m, 3H), 5.12–5.00 (m, 1H), 3.35 (dd, *J* = 17.5, 10.8, 1H), 3.17–3.04 (m, 2H), 2.97–2.87 (m, 1H), 2.36 (s, 3H); ^13^C NMR (101 MHz, CDCl3) δ 186.6, 157.8, 138.4, 136.1, 135.9, 133.7, 130.4(2C), 130.4, 128.7, 128.5(2C), 127.9, 126.6, 83.8, 40.9, 38.5, 21.5; IR νmax 3048, 2908, 1661, 1581, 862, 750, 691 cm^−1^; HRMS (EI) *m*/*z* calcd for C_18_H_18_NO_2_ [M + H]^+^ 280.1332, found 280.1340.

#### 3.2.12. (5-(4-Methylbenzyl)-4,5-dihydroisoxazol-3-yl)(phenyl)methanone (**5c**) 

The compound was prepared following the general procedure. Yield 64%. ^1^H NMR (400 MHz, CDCl_3_) δ 8.16–8.10 (m, 2H), 7.59 (m, 1H), 7.46 (m, 2H), 7.16 (m, 4H), 5.05 (dd, *J* = 11.0, 7.8 Hz, 1H), 3.35 (dd, *J* = 17.6, 10.8 Hz, 1H), 3.15–3.04 (m, 2H), 2.93 (dd, *J* = 14.0, 6.6 Hz, 1H), 2.34 (s, 3H); ^13^C NMR (101 MHz, CDCl3) δ 186.7, 157.9, 136.8, 136.1, 133.8, 133.2, 130.5(2C), 129.6(2C), 129.6(2C), 128.6(2C), 83.9, 40.7, 38.5, 21.3; IR νmax 3039, 2909, 1710, 1609, 822, 741, 680 cm^−1^; HRMS (EI) *m*/*z* calcd for C_18_H_18_NO_2_ [M + H]^+^ 280.1332, found 280.1339.

#### 3.2.13. (5-(4-Methoxybenzyl)-4,5-dihydroisoxazol-3-yl)(phenyl)methanone (**5d**) 

The compound was prepared following the general procedure. Yield 72%. ^1^H NMR (400 MHz, CDCl_3_) δ 8.12 (m, 2H), 7.58 (m, 1H), 7.45 (m, 2H), 7.18 (m, 2H), 6.86 (m, 2H), 5.08–4.96 (m, 1H), 3.78 (s, 3H), 3.34 (dd, *J* = 17.6, 10.8 Hz, 1H), 3.14–2.99 (m, 2H), 2.91 (dd, *J* = 14.1, 6.4 Hz, 1H); ^13^C NMR (101 MHz, CDCl3) δ 186.6, 158.7, 157.8, 135.9, 133.6, 130.6(2C), 130.4(2C), 128.4(2C), 128.1, 114.2(2C), 83.8, 55.3, 40.0, 38.3; IR νmax 3050, 2850, 1670, 1580, 820, 750, 691 cm^−1^; HRMS (EI) *m*/*z* calcd for C_18_H_18_NO_3_ [M + H]^+^ 296.1281, found 296.1285.

#### 3.2.14. (5-Octyl-4,5-dihydroisoxazol-3-yl)(phenyl)methanone (**5e**)

The compound was prepared following the general procedure. Yield 89%. ^1^H NMR (400 MHz, CDCl_3_) δ 8.25–8.12 (m, 2H), 7.58 (m, 1H), 7.46 (m, 2H), 4.79 (ddt, *J* = 10.9, 8.4, 6.6 Hz, 1H), 3.39 (dd, *J* = 17.4, 10.9 Hz, 1H), 3.00 (dd, *J* = 17.4, 8.5 Hz, 1H), 1.79 (m, 1H), 1.69–1.57 (m, 1H), 1.45–1.19 (m, 12H), 0.88 (m, 3H); ^13^C NMR (101 MHz, CDCl3) δ 186.7, 157.9, 136.0, 133.6, 130.5(2C), 128.5(2C), 83.7, 38.9, 35.3, 31.9, 29.6, 29.5, 29.3, 25.4, 22.8, 14.2; IR νmax 3062, 2948, 1635, 1541, 760, 710 cm^−1^; HRMS (EI) *m*/*z* calcd for C_18_H_26_NO_2_ [M + H]^+^ 288.1958, found 288.1961. These data are consistent with the data reported in the literature [11].

#### 3.2.15. (5-(Chloromethyl)-4,5-dihydroisoxazol-3-yl)(phenyl)methanone (**5f**) 

The compound was prepared following the general procedure. Yield 83%. ^1^H NMR (400 MHz, CDCl_3_) δ 8.24–8.14 (m, 2H), 7.66–7.55 (m, 1H), 7.53–7.44 (m, 2H), 5.05 (dddd, *J* = 11.1, 7.1, 5.7, 4.5 Hz, 1H), 3.76–3.62 (m, 2H), 3.50 (dd, *J* = 17.9, 11.1 Hz, 1H), 3.36 (dd, *J* = 17.9, 7.1 Hz, 1H); ^13^C NMR (101 MHz, CDCl3) δ 186.0, 157.5, 135.7, 133.9, 130.5(2C), 128.6(2C), 81.2, 45.1, 37.7; IR νmax 3025, 1660, 1580, 700 cm^−m^; HRMS (EI) *m*/*z* calcd for C_11_H_11_NO_2_Cl [M + H]^+^ 224.0473, found 224.0466. These data are consistent with the data reported in the literature [11].

#### 3.2.16. (3a,4,5,6,7,7a-Hexahydrobenzo[d]isoxazol-3-yl)(phenyl)methanone (**5g**) 

The compound was prepared following the general procedure. Yield 54%. ^1^H NMR (400 MHz, CDCl_3_) δ 8.18 (m, 2H), 7.61–7.56 (m, 1H), 7.47 (m, 2H), 4.59 (dt, *J* = 7.9, 3.9 Hz, 1H), 3.46–3.38 (m, 1H), 2.20 (dd, *J* = 15.2, 3.5 Hz, 1H), 2.11–2.01 (m, 1H), 1.82 (tt, *J* = 15.3, 4.7 Hz, 1H), 1.65–1.52 (m, 3H), 1.35–1.24 (m, 2H); ^13^C NMR (101 MHz, CDCl3) δ 187.0, 163.8, 136.4, 133.6, 130.4(2C), 128.5(2C), 82.3, 44.3, 25.6, 25.0, 21.7, 19.9; IR νmax 3060, 2940, 1660, 1550, 747, 704 cm^−m^; HRMS (EI) *m*/*z* calcd for C_14_H_16_NO_2_ [M + H]^+^ 230.1176, found 230.1182. These data are consistent with the data reported in the literature [11].

#### 3.2.17. Ethyl 3-benzoyl-4,5-dihydroisoxazole-5-carboxylate (**7a**)

The compound was prepared following the general procedure. Yield 68%. ^1^H NMR (400 MHz, CDCl_3_) δ 8.33–8.28 (m, 2H), 7.71–7.65 (m, 1H), 7.59–7.51 (m, 2H), 7.43 (s, 1H), 4.48 (q, *J* = 7.1 Hz, 2H), 1.44 (t, *J* = 7.1 Hz, 3H); ^13^C NMR (101 MHz, CDCl3) δ 184.7, 162.3, 161.3, 156.4, 135.3, 134.6, 130.9(2C), 128.9(2C), 110.2, 62.8, 14.3; IR νmax 3058, 2945, 1655, 1545, 740, 702 cm^−m^; HRMS (EI) *m*/*z* calcd for C_13_H_14_NO_2_ [M + H]^+^ 248.0917, found 248.0912.

#### 3.2.18. (5-Butylisoxazol-3-yl)(phenyl)methanone (**7b**) 

The compound was prepared following the general procedure. Yield 64%. ^1^H NMR (400 MHz, CDCl_3_) δ 8.29 (m, 2H), 7.61 (m, 1H), 7.49 (m, 2H), 6.51 (ms, 1H), 2.82 (t, *J* = 7.6 Hz, 2H), 1.73 (dt, *J* = 15.2, 7.5 Hz, 2H), 1.42 (dq, *J* = 14.6, 7.4 Hz, 2H), 0.95 (t, *J* = 7.4 Hz, 3H); ^13^C NMR (101 MHz, CDCl3) δ 186.2, 174.8, 161.9, 135.9, 133.9, 130.7(2C), 128.6(2C), 101.7, 29.6, 26.4, 22.3, 13.8. IR νmax 3075, 2950, 2875, 1670, 1590, 740, 690 cm^−m^; HRMS (EI) *m*/*z* calcd for C_14_H_16_NO_2_ [M + H]^+^ 230.1176, found 230.1171. These data are consistent with the data reported in the literature [11].

## 4. Conclusions

Isoxazolines and isoxazoles are biologically active molecules. The development and improvement of syntheses directed towards isoxazolines and isoxazoles is a continuing pursuit. Herein, we have developed an effective cycloaddition of various α-nitroketones with alkenes or alkynes by using the cheap base chloramine-T. The low cost and ease of handling of this moderate base are its outstanding properties. The cycloaddition described in this study is an integrated approach for synthesizing isoxazolines and isoxazoles. We are currently investigating other ways to integrate isoxazolines. 

## Data Availability

The data presented in this study are available in the article and Appendix A.

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
