# Peer review of "3-Benzoylisoxazolines by 1,3-Dipolar Cycloaddition: Chloramine-T-Catalyzed Condensation of α-Nitroketones with Dipolarophiles"

_molecules, 2021, doi:10.3390/molecules26123491_

Round 1
Reviewer 1 Report
The manuscript of Pan &all is dealing with the Chloramine-T catalytic proprieties in condensation of α-Nitroketones with Dipolarphiles. The chemistry is simple and seems to be efficient. However, there are some unclear aspects:
- The authors are using acetonirile as solvent in the experimental setup. Acetonirile have a moderate toxicity. In Conclusions (pag 10) they claim that their experimental setup is non-toxic. I disagree, they have to change. Has to be rewritten;
-Abstract: it provides general data which is unusual. Has to be rewritten
- The English is clumsy, minor spell check required (eg pag.2, line 33 is alfa-nitrok not anitroketones.
Reviewer 2 Report
This paper reports the use of chloramine T to promote the reaction of alpha-nitroketones and alkenes (alkynes) to form isoxazolines (isoxazoles). The use of chloramine T in the preparation of isoxazolines from nitro compounds would appear to be novel and good yields are reported. However, chloramine T has been widely used in the generation of nitrile oxides from oximes and the subsequent formation of isoxazolines by 1,3-dipolar reactions with alkenes. Chloroamine T is usually used as on oxidising agent. It is only a weak base (pKa ca. 9.5).
So as this paper reports new conditions for the formation of isoxazolines from nitroketones and alkenes it can be accepted for publication. However, extensive revision is required.
A mechanism is proposed but this is purely speculative as no evidence in support of it is described. Moreover parts of the mechanism simply do not make sense. For example, the arrows drawn for the elimination of the isoxazoline from the complex on the bottom left hand corner of the mechanistic scheme are simply wrong and don't lead to the outcome predicted. Moreover the proposed cycloaddition step involves the addition of a deprotonated nitroketone, albeit hydrogen bonded to the protonated chloramine T, to an electron rich alkene. This is unprecedented and wrong.
The mechanism as drawn must be deleted.
The formation of isoxazolines from nitro compounds and alkenes usually involves dehydration of the nitro compound to form a nitrile oxide that undergoes a dipolar cycloaddition reaction with the alkene. In the absence on an alkene, nitrile oxides dimerise to form furoxans. In the absence of alkenes are furoxans formed from the nitroketones and chloroamine T?
Does the reaction work for simple nitrocompounds or is it limited to nitroketones?
Chloramine T is only a weak base but it probably can still deprononate the nitroketones. The question is then whether the chlorosulfonamide can dehydrate the deprotonated nitroketone to generate a nitrile oxide. My guess is that it can but I would need to search the literature for analogous reactions first.
The English needs significant improvement throughout the paper. The 1H NMR spectrum for 3a is interpreted incorrectly since it cannot have a dt with a coupling constant of 21.5 Hz for 5 hydrogens. I haven't checked the other 1H data.
References should be given in the experimental to all known compounds - that is if any more are known apart from 3a.
To me a reagent that is used in an amount of 0.5 equiv. and not recovered is not a catalyst. It should be mentioned that a large excess of the alkene is required. This is not uncommon for cycloaddition reactions of nitrile oxides.
Round 2
Reviewer 2 Report
This paper describes a new reaction that should be published but as drawn the mechanism is still flawed. The authors cite a reference in the literature but either this is flawed as well or it has been interpreted incorrectly.
in this reaction, either the nitroketone is being dehydrated to form an nitrile oxide (alpha to the ketone) which add to the alkenes to form the isoxazolines or the nitroketone is being deprotonated to form a nitronate anion that adds to the alkene to give a 2-hydroxyoxazolidine anion that is then protonated to give the 2-hydroxyoxazolidine that is dehydrated to give the product. The authors prefer the latter process. Although anionic species usually do not add to electron rich alkenes, the authors propose the participation of complex hydrogen bonded intermediates. However, the mechanism is drawn incorrectly. For example the addition of the nitronate to the alkene, that has a vinyl hydrogen on both carbons of the double bond, cannot give rise to the intermediate given as the product of this reaction that has undergone a further deprotonation. This must be a separate step. A simplified mechanism that has plausible arrows is attached.
The grammar needs further improvement

Author Response
Please see the attachment.

This manuscript is a resubmission of an earlier submission. The following is a list of the peer review reports and author responses from that submission.